# Functional Food Components, Intestinal Permeability and  Inflammatory Markers in Patients with Inflammatory Bowel Disease

**DOI:** 10.3390/nu13020642

**Published:** 2021-02-16

**Authors:** Joana Franco Lacerda, Ana Catarina Lagos, Elisabete Carolino, Ana Santos Silva-Herdade, Manuel Silva, Catarina Sousa Guerreiro

**Affiliations:** 1Nutrition Laboratory, Faculty of Medicine, University of Lisbon, 1649-045 Lisbon, Portugal; joanalebrelacerda@gmail.com; 2Hospital of Armed Forces Lisbon Pole, 1649-020 Lisbon, Portugal; catarinalagos@gmail.com (A.C.L.); marsilva@hfar.pt (M.S.); 3H&TRC—Health & Technology Research Center, Polytechnic Institute of Lisbon, School of Health Technology, 1990-096 Lisbon, Portugal; etcarolino@estesl.ipl.pt; 4Faculty of Medicine University of Lisbon, Institute of Molecular Medicine, University of Lisbon, 1649-045 Lisbon, Portugal; anarmsilva@medicina.ulisboa.pt; 5Faculty of Medicine, Institute of Environmental Health, University of Lisbon, 1649-026 Lisbon, Portugal

**Keywords:** inflammatory bowel disease, intestinal permeability, serum zonulin, inflammation, functional foods

## Abstract

Inflammatory bowel diseases (IBD) are characterized by a chronic inflammatory process that affects the intestinal barrier structure. Recent evidence suggests that some food components can influence the integrity of the intestinal barrier and thus its permeability. We aimed at assessing the effect of food components on the intestinal permeability (IP) and on inflammatory markers in individuals with IBD by a single-blind randomized clinical study. Of the 53 individuals included, 47% (*n* = 25) had been diagnosed with IBD. The participants were divided into 4 groups. IBD patients were allocated to intervention group (*n* = 14) vs. no intervention group (*n* = 11), and the same happened with 28 control participants without disease (*n* = 14 in intervention group vs. *n* = 14 without intervention). Symptomatology, nutritional status, biochemical parameters (specifically serum zonulin (ZO) to measure IP) were evaluated on all individuals on an eight week period following a diet plan with/without potentially beneficial foods for the IP. At the beginning of the study, there were no significant differences in ZO values between individuals with and without IBD (*p* > 0.05). The effect of specific food components was inconclusive; however, a trend in the reduction of inflammatory parameters and on the prevalence of gastrointestinal symptomatology was observed. More controlled intervention studies with diet plans, including food components potentially beneficial for the integrity of the intestinal barrier, are of the utmost importance.

## 1. Introduction

Inflammatory bowel disease (IBD) is characterized by a chronic inflammatory process of the gastrointestinal tract and is marked by a disruption of the intestinal barrier that influences intestinal permeability (IP). Although its pathogenesis is not fully understood, recent evidence suggests that IBD is associated with a multifactorial process involving genetics, environmental factors, microbiota, and deregulation of the immune system [1].

The intestinal barrier is composed of epithelial cells that are united by tight junction proteins (TJP). These have a crucial role on intestinal permeability by conferring selectivity on the ion flux, small molecules and solutes [2]. Recent studies have signaled serum zonulin (ZO) as a marker of IP with a high potential for clinical use in various pathologies in adults and children [3]. The concentration of serum ZO rises when the IP is high, presenting a direct correlation with the IP urine test (quotient mannitol lactulose) [4,5,6]. Due to the dynamic structure of the TJP bonds, the intestinal barrier permeability will suffer changes caused by biological factors and components from the intestinal mucosa as well as by external factors such as nutrients and their metabolites [7].

Different in vitro studies have previously evaluated the effect of specific food components on IP by measuring the transepithelial electric resistance that measures the ion flux between cells [7,8]. Components having a positive effect include polyphenols (quercetin and curcumin), tryptophan, zinc, vitamin D, prebiotics (beta-glucans and butyrate) and glutamine [1,7,9,10,11,12,13,14,15,16,17,18,19,20,21,22]. These components revealed a protective role on the integrity of TJP from the intestinal barrier, thereby reducing its permeability:-Curcumin and quercetin are two polyphenols that occur in nature and can influence the intestinal barrier structure. It has already been demonstrated that curcumin can block tumor necrosis factor α (TNFα) and interleukin-1β (IL-1β) induced nuclear factor kappa B (NF -κB) activation, increasing the transepithelial electric resistance and reducing IP [7]. The polyphenol quercetin is the most common flavonoid in nature, and in vitro studies verified that it increases the transepithelial electric resistance and reduces paracellular flux by an increase in claudin-4 and the assembly of ZO-2, occludin and claudin-1 at the TJP level [7].-Tryptophan: can reduce the IP and improve the TJP function, especially by raising the level of ZO-1 [9], as well as its metabolites can lead to changes in the microbiota that enhance the protection against gastrointestinal infections [19].-Zinc: it seems to present a crucial role in the intestinal barrier integrity through keeping the expression of occludin and claudin-3 and suppressing the proteolysis of occludin [12]. Finamore et al. verified in vitro the importance of zinc in the maintenance of membrane barrier function and in controlling inflammatory reactions. The depletion of zinc caused changes in TJP, promoting the migration of neutrophils [18].-Vitamin D: the biological activity of 1,25-dihydroxyvitamin D is mediated by the vitamin D receptor that is highly expressed in gut epithelial cells, and its signaling seems to preserve the integrity of the intestinal barrier by inhibiting the phosphorylation of myosin, as well as by raising the expression of TJP [7,14,21].-Prebiotics (beta-glucans and butyrate): it was verified, in vitro, that the increase of dietary fiber stimulates the production of mucus and the expression of TJP, leading to a reduction of the IP [10,11]. In rats, butyrate strengthens the intestinal barrier through the increase of transepithelial electric resistance [7]. The β-glucans fiber, as it is fermented, leads to a rise in fecal butyrate concentration [20].-Glutamine: is the main source of amino acids for the intestinal mucosa and can restore stress-induced loss of barrier integrity by increasing transepithelial electric resistance [7]. Despite the importance of glutamine in the small intestine, excessive glutamine in the colon leads to increase oxidative tissue injury; thus, its supplementation is controversial [22].

On the other hand, components such as gliadin, ethanol, sugar, fat and food additives have shown to have a damaging effect on intestinal barrier integrity. They have various harmful effects, namely: reduction of butyrate production and the thickness of mucus layer, damages in epithelial cells, raise of Gram-negative bacteria, reduction of transepithelial electric resistance and TJP disorganization leading to a rise of IP [7,8,23].

In this study, we evaluated the effect of a nutritional intervention on IP and inflammatory markers in individuals with IBD. We also evaluated the impact of the inclusion of potentially beneficial foods for the IP and the inflammatory profile of the gastrointestinal symptomatology of individuals with IBD.

## 2. Materials and Methods

### 2.1. Participants

This single-blind randomized clinical study was approved by the Ethics Commission of the Centro Hospitalar Lisboa Ocidental and of Hospital das Forças Armadas—Pólo de Lisboa (HFAR-PL).

The study was composed of 2 groups that included HFAR-PL outpatients. All subjects gave their informed written consent for inclusion before they participated in the study. The study was conducted in accordance with the Declaration of Helsinki.


-Patients group (*n* = 25), in which we established as inclusion criteria the diagnosis of IBD (after clinical, endoscopic and histological evaluation) and age equal or superior to 18. The exclusion criteria were pregnancy/breastfeeding, rheumatoid arthritis, asthma, diabetes mellitus type 1 (DM1), celiac disease, human immunodeficiency virus (HIV), irritable bowel syndrome (IBS), hepatic steatosis, dyspepsia and body mass index (BMI) equal or superior to 30 kg/m^2^;-Control group (*n* = 28), with age equal or superior to 18 as inclusion criteria. As exclusion criteria, we considered: gastrointestinal pathologies, pregnancy/breastfeeding, rheumatoid arthritis, asthma, DM1, HIV and BMI equal or superior to 30 kg/m^2^


The patient group was identified and selected based on the defined inclusion and exclusion criteria by the gastroenterologist. As for the control group, participants were identified and selected by the gastroenterologist in routine consultation and by the nutritionist in general nutrition consultation.

A simple randomization was applied. Both groups (control and cases) were allocated randomly into four groups: patient group with intervention (*n* = 14); patient group without intervention (*n* = 11); control group with intervention (*n* = 14), and control group without intervention (*n* = 14).

Prior to intervention, all participants attended their initial nutrition appointment, where the following measurements were obtained: hemogram, c-reactive protein (CRP), as well as iron, folic acid, vitamin D and B12, zinc, calcium, zonulin and fecal (calprotectin) analysis.

The participants in this study were accompanied and evaluated twice in person (at study initiation and completion) as well as monitored by telephone contact after 1 month of the first nutrition appointment.

### 2.2. Study Design and Methodology

The complete study design is illustrated in Figure 1 and described below.

#### Initial Evaluation (M1)

During the first nutrition appointment (M1), the following sociodemographic and clinical data were collected: IBD duration, extension and activity, pharmacotherapy, vitamins or mineral supplements and former gastrointestinal tract surgery. We also evaluated the gastrointestinal symptoms through an analogical scale in order to evaluate their severity during the last four weeks, namely: abdominal pain and distension, constipation, diarrhea, flatulence and heartburn. There was also an evaluation of the body composition by bioimpedance (electric bioimpedance scale Tanita SC-330) and food report by the 24 h Recall method.

In the end, a personalized diet plan (DP) adjusted to their energy needs, based on the Mediterranean diet (diet plan composed of 7 meals including vegetables, fresh fruit, nuts, cereals and grains, fish, eggs, dairy produce that favors the consumption of unsaturated fats such as olive oil instead of saturated fats) was given to all participants. Only for the intervention groups, foods and nutrients as curcumin, tryptophan, zinc, vitamin D, quercetin and beta-glucans were included in the plan, and their intake was strongly encouraged (the DP mentioned, for each meal, the quantity and frequency of the daily intake of foods rich on those nutrients, as well as the importance of their intake during the first appointment—M1). By opposite alcohol and gliadin, intake was reduced. The participants from the intervention groups also received other nutritional advice, aiming the inclusion of food rich in glutamine and the exclusion of food rich in fat, sugar and additives as E407 (carrageenan) and E466 (cabroximetilcelulosis). The DP for the intervention groups had the following information:-The intake of gliadin was limited to 4 times per week and only on one of the 7 meals mentioned in the DP;-Alcohol intake limited to a maximum one dose (100–140 mL) intake per week;-All food with a high content of lipids and/or sugar were limited to a max intake of 1 to 2 times a month: all foods with more than 17.5 g of lipids or more than 5 g of saturated fat by 100 g of feed and more than 8.75 g of lipids or more than 2.5 g of saturated fat per 100 mL of beverages and/or more than 22.5 g of sugar per 100 g of feed or more than 11.25 g of sugar per 100 mL of beverage;-Foods having in their composition additives E407 (carrageenan) and E466 (cabroximetilcelulosis) were prohibited.

As for curcumin, we specified a daily intake of 2 g/day and provided each participant with the necessary doses for all study duration. No food supplements or pills were included; only participants that presented values of vitamin D lower than 30 ng/mL started oral supplementation.

All participants were asked to carry out the DP for eight weeks.

One-month follow-up (M2) The second moment (M2) occurred one month after M1. It consisted of telephone contact to all participants in order to evaluate the fulfillment of the DP and to allow the participants to clarify any doubts.

Final evaluation after two months (M3) At the final moment (M3), the same evaluations that occurred at M1 were done, and also participants answered a qualitative and multiple-choice questionnaire regarding satisfaction and rate of adhesion to DP. It had questions regarding their fulfillment of doses and frequency for each food/nutrient/food additive with the potential effect on the intestinal permeability.

Figure 1 shortly details the design and methodology of this study.

### 2.3. Statistical Analysis

Data were analyzed using IBM Statistical Package for Social Science software (version 22.0, IBM Chicago, IL, USA). The differences between quantitative variables of two independent samples were evaluated using Student’s *t*-test or the Mann–Whitney test, depending on data normality. In order to compare the first and the last moments of the evaluation of the same group, the Student’s *t*-test for two matched samples or Wilcoxon test was used. The association between dichotomic variables was evaluated using Fisher’s exact test, whereas the association between two qualitative variables was analyzed through the Chi-square test by the Monte Carlo simulation. The correlation between quantitative variables was made using Spearman’s rank correlation coefficient test. A 95% confidence interval was utilized for significance (*p* < 0.05 was considered significant)

## 3. Results

### 3.1. Participant Characteristics

A total of 53 individuals were included in this study, as described in Table 1. From those, 25 belonged to the patient group and 28 from the control group. During the study, two individuals from the control group withdrew. The minimum age of the participants was 26, and the maximum was 75. No differences on age (*p* = 0.215) or gender (*p* = 1.0) were observed between groups. Approximately 18% of individuals from the control group and only 12% of individuals from the case group had smoking habits.

Regarding the clinical data, no comorbidities were found in almost 70% of IBD patients IBD and 50% of controls. Furthermore, the majority of the participants did not intake any vitamins nor mineral supplementation (96.4% in the control group and 76% in the cases group).

### 3.2. Medication and Disease Activity Evaluation

In the beginning, only 1/25 participants with IBD had no medication. The majority was taking 5-ASA (5-aminosalicylic acid). During the study, 1 participant raised the medication dosage, 6 reduced, 1 stopped corticotherapy, and 5 reduced the dosage of 5-ASA. In the end, all participants were under some kind of therapeutic.

The majority of the participants were in remission before the study began. Only 3 with ulcerative colitis (UC) presented mild activity that was kept until the end of the study, and 1 (UC) that started in remission ended the study with moderate activity (Table 2.).

### 3.3. Anthropometric Evaluation

The results of the anthropometric evaluation are shown in Table 2. A significant reduction (*p* < 0.05) in weight, BMI, waist circumference and body fat between moment 1 and 3 was observed in both intervention groups (IBD cases and controls) (Table 3.).

### 3.4. Nutritional Intake

At M1, comparing IBD patients with controls, no significant differences were seen in average daily energy intake as well as in the majority of different macro and micronutrients (*p* > 0.05). Nevertheless, significant higher intake of protein (*p* = 0.04), alcohol (*p* = 0.01), calcium (*p* = 0.02), zinc (*p* = 0.04) and tryptophan (*p* = 0.04) was seen in controls (data not shown). After 2 months intervention, a reduction in lipids (*p* = 0.05), iron (*p* = 0.02) and an increase in fiber intake (*p* = 0.01) were observed in IBD intervention group. At control intervention group the same happened with iron, but also with alcohol intake. (*p* = 0.01). (Table 4.)

### 3.5. Food Patterns and Restrictions

Our results revealed that the IBD group tends to do more food restrictions comparing with individuals without the disease, namely: restriction in dairy products, meat/fish, fat, cereals, vegetables, and alcohol.

### 3.6. Biochemical Evaluation

The results of the biochemical evaluation are presented in Table 5. At the beginning of the study, participants diagnosed with IBD presented as expected significantly higher CRP and fecal calprotectin values. It was also verified that IBD individuals displayed a significantly lower ferritin concentration when compared with the control group (*p* < 0.05).

When comparing the biochemical parameters between the different moments, we observed a decrease in CRP values in all groups, albeit not significant; this difference was higher in the intervention groups (*p* > 0.05). A non-significant decline of fecal calprotectin was observed in all groups. For ZO, a non-significant difference was verified between moments in all groups (Table 5).

### 3.7. Gastrointestinal Symptomatology

Regarding gastrointestinal symptomatology evolution, non-significant changes from M1 to M3 were observed (*p* > 0.05), although a decrease of several gastrointestinal symptoms’ prevalence was observed (Table 6.)

### 3.8. Relation between Values of Serum Zonulin and Gastrointestinal Symptomatology, Anthropometric Evaluation and Clinical Variables

At the initial consultation, a significant correlation was observed between serum ZO and the following variables: abdominal distension (*r* = 0.282, *p* = 0.04), CRP (*r* = 0.375, *p* = 0.006), fecal calprotectin (*r* = 0.343, *p* = 0.004), BMI (*r* = 0.346, *p* = 0.01) and waist circumference (*r* = 0.272, *p* = 0.049). These correlations lost their clinically relevance (*p* > 0.05) when only IBD individuals were studied.

It was also noted that IBD individuals taking immunosuppressor medication tended to show lower values of ZO (48.7 ± 21.3 vs. 50.9 ± 14.4) (*p* = 0.06).

Moreover, higher values of ZO were observed in individuals in which CD only affects the colon and in UC individuals with the full length of the colon affected (*p* > 0.05).

No association between ZO values and years of disease, disease activity and the fulfillment of the diet plan were verified in individuals with IBD.

### 3.9. Diet Plan Adhesion and Satisfaction

According to patients reports, almost 60% (9 cases and 7 controls) of the individuals from the intervention groups met more than 50% of the plan recommendations, whereas only 7.4% (0 cases and 2 controls) fulfilled more than 90% and 3.7% (0 cases and 1 control) met 100% of the prescribed DP.

We asked participants about nutritional plan satisfaction considering a 1 to 10 scale. 85.2% (13 cases and 10 controls) classified satisfaction equal to or higher than 8.

## 4. Discussion

The results of this exploratory study showed that our nutritional intervention promoted a significant reduction in patient’s body weight, with a lower lipid and a higher fiber intake.

Despite both intervention groups mentioning a high satisfaction with the DP, especially in the IBD group, the plan adherence fell short from the desired, as none of the IBD individuals managed to reach a 90% rate of fulfillment, leading to the doubt whether the improvements observed were associated with DP or due to other variables like therapeutics.

Accordingly, with the literature, changes of the intestinal barrier that characterizes IBD leads to a rise of the IP that can be verified through the rise of serum ZO concentration [4,24,25,26]. In asymptomatic CD individuals, the rise of IP precedes the active phase of the disease, which may happen until one year before the relapse. Such evidence is also supported when verifying that first-grade relatives of UC and CD individuals, clinically asymptomatic, can also show a risen PI [24,25,26]. Alessio Fasano highlights that the rise of ZO is detectable in the active phase of IBD and that its serum levels lower in the remission phase, although higher than the levels of healthy individuals [27].

Contrary to what has been described in the above-mentioned literature, in our study, the initial values of ZO were not significantly different between IBD and controls [4,23,28,29,30,31,32,33].

Analyzing all the samples at the initial moment, a positive correlation was verified between serum ZO and CRP, fecal calprotectin, BMI, waist circumference, abdominal distention. Although contrarily to what Calviglia et al. observed when only analyzing our IBD individuals, at the initial moment, they did not present a correlation between serum ZO concentration and duration of the disease, neither with BMI, CRP, fecal calprotectin, disease activity and DP adherence. The correlation observed by us regarding abdominal distension and ZO was not referred by any of the authors in previous studies.

In this study, as in others [4,5,6,34], ZO was used as an IP marker as it facilitates the individuals’ participation. In the literature, there are few authors that, by relating specific food components with changes of IP, indicate the doses needed to be effective. When they do, doses are significantly different between studies. One of the most referred and quantified components is curcumin, although the lack of consensus on the ideal dosing is evident [1,35]. In this study, a 2 g/day dose was used, the same as Hanai et al., who associated this intake together with 5-ASA to higher efficiency in maintaining UC remission when compared with placebo and 5-ASA.

It is interesting that in this study, just as in Buning et al.’s study, a relation between the type of medication and serum ZO values were registered. Although being a not significant difference, the individuals with an immunosuppressor therapeutic showed lower serum ZO values when compared with individuals without this type of medication.

As previously mentioned, a possible effect of the disease extension on the serum values of ZO was observed. In individuals that CD only affects the colon and in those that UC affects the full length of the colon; the values of ZO were higher. Buning et al. showed the same, but only on UC individuals.

We may thus assume that the rise of IP is directly related to circulant cytokine concentration [27]. This argument could justify the rise of IP in individuals with pancolitis when compared to individuals with proctitis. However, Buning et al. highlight that in their study, the fecal calprotectin values do not relate with the disease extension (UC), thus maintaining the question regarding the relationship between the extension of IBD and IP [24].

Wegh et al. showed that individuals with UC in clinical remission did not show a risen IP [36]. In our study, as in Calviglia et al., no correlation between the serum ZO concentration and the disease activity was verified. Other authors argue that the changes of TJP and IP in individuals with IBD are maintained even in the absence of inflammation/during the remission phase. They also argue that asymptomatic first-degree relatives of individuals with IBD show significant rises of IP, claiming that the changes that occurred in the intestinal barrier are genetic, thus being constant, regardless of the disease activity [2,24,25,26,37,38].

The true relation between the disease activity and IP needs to be clarified.

Serum ZO concentration has been questioned by some authors regarding its sensibility for evaluating IP. Ajamian et al. argued that the current commercialized tests (imunoenzyme trials) do not have the sufficient sensibility to differentiate ZO from other proteins [39]. Ohlsson et al. also mentioned that, as there are more than 50 proteins involved in the TJP and IP regulation, the use of ZO as an isolated marker of IP should be reconsidered. Thoughts that contradict Mokkala et al. and Sapone et al. that argue the existence of a direct correlation between sugar tests in urine and the serum concentration of ZO [5,6].

For now, there is no agreement regarding the use of ZO as an IP marker, particularly when referring to its role along gastrointestinal tract location. In contrast with the majority of the studies considering that ZO acts along all the gastrointestinal tract, recent studies suggest that ZO has its action limited in the small bowel, arguing that it should only be used as an IP marker in individuals with that part of the bowel affected [4,36,40].

The relation of ZO and metabolic diseases has been previously described, and our study confirmed that association, as the ZO levels were positively related to BMI and waist circumference [41].

Our research had, as main limitations, the reduced sample size and its short duration. Nevertheless, albeit the reduced sample, the presence of four study arms was extremely important.

We also need to mention that specific food components added to DP of intervention groups could not be adequately quantified, namely curcumin, beta-glucans, glutamine, polyphenols and food additives. This is due to the absence of a database appropriated to the Portuguese population.

When comparing with other recent studies, our work stands out as one of the first intervention studies where a DP that includes food components with potential theoretical benefits for the integrity of the intestinal barrier was applied.

In order to improve future studies, it needs to be taken into account necessary to ensure a better fulfillment of the established DP.

## 5. Conclusions

In conclusion, our results show that the benefit in including and excluding, in short-term, specific food components, potentially beneficial or harmful for the IP of IBD patients is inconclusive. Therefore, it is still uncertain if IBD individuals could benefit from this type of nutritional therapeutic. The intake/restriction of those nutrients/food did not affect, in a significant way, the IP, the inflammatory parameters, nor the gastrointestinal symptomatology. It is necessary to confirm and deepen the reliability of ZO as an IP marker.

More studies are needed, with larger samples, in a more controlled environment, using gold-standard tests to measure the IP and probably adding microbiota analysis in order to evaluate the real interest in promoting food pattern changes in this type of patient.

Although this study was inconclusive, more robust studies have emerged demonstrating the scientific rationale in the benefits of specific food for intestinal health. The same happens when referring to the association in the rise of IP and IBD. Having this in mind, continuing and developing more studies in this area is of the utmost importance.

## Figures and Tables

**Figure 1 nutrients-13-00642-f001:**
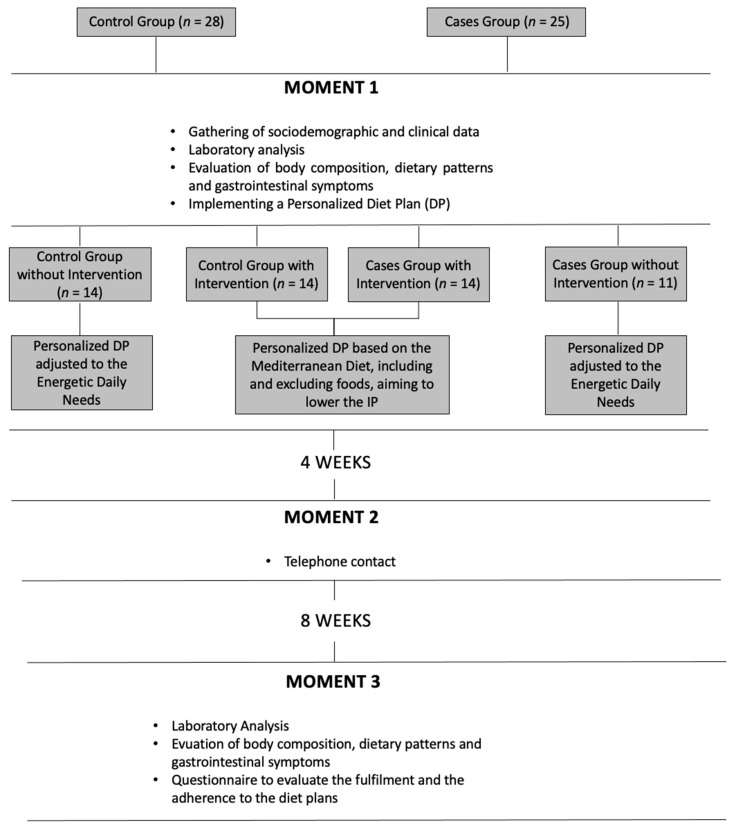
Study design and methodology.

**Table 1 nutrients-13-00642-t001:** General participant characteristics.

General Participant Characteristics		Cases *N* (%)				Controls *N* (%)				*p* Value
Gender	Female	5 (20)				5 (17.9)				1.000
	Male	20 (80)				23 (82.1)				
Smoking habits	Yes	3 (12)				5 (17.9)				0.708
	No	23 (88)				23 (82.1)				
		Cases				Controls				
		Average (±SD)	Median	IQR (Q25%–Q75%)	Range	Average (±SD)	Median	IQR (Q25%–Q75%)	Range	*p* Value
Age		44.04 (±12.3)	44	19.5 (34–53.5)	47	48.32 (±12.4)	51.5	19.25 (37.5–56.8)	49	0.215
Weight (kg)		77.9 (±11.1)	78.5	12.95 (73.6–86.6)	44.3	75.4 (±8.8)	75.1	15 (68.4–83.3)	33.4	0.359
BMI (kg/m^2^)		25.8 (±2.9)	25.3	5.35 (23.2–28.6)	10.4	25.5 (±2.2)	25.1	3.3 (23.7–27)	7.3	0.640
Waist Circumference (cm)	Female	78.6 (±10.4)	72	17.5 (71.5–89)	24	78.6 (±10.1)	76	16.5 (71–87.5)	26	1.000
	Male	94.1 (±8.9)	94.5	12.8 (89–101.8)	33	90.7 (±7.1)	90	10 (85–95)	27	0.165
% Body Fat	Female	31.9 (±8.6)	28.6	16.35 (24.6–40.9)	20	31.74 (±6.3)	31.9	11.4 (26–37.4)	16.3	0.974
	Male	21.5 (±3.9)	21.1	5.8 (19–24.8)	15.3	20.13 (±4.2)	20.2	4.5 (17.8–22.3)	20.2	0.294

SD: standard deviation; BMI: body mass index; IQR: interquartile range.

**Table 2 nutrients-13-00642-t002:** Medication and activity disease on individuals with IBD.

IBD	Medication and Activity Disease	CD (*N* = 13) *N* (%)		UC (*N* = 12) *N* (%)	
		M1	M3	M1	M3
Medication	Without medication	1 (7.7)	0	0	0
	5-ASA	4 (30.8)	4 (30.8)	6 (50)	6 (50)
	Systemic steroids	2 (15.4)	2 (15.4)	1	0
	Anti-TNF-α	1 (7.7)	1 (7.7)	0	0
	5-ASA + immunosuppressors	2 (15.4)	3 (23,1)	2 (16.7)	2 (16.7)
	5-ASA + systemic steroids	0	0	1 (8.3)	1 (8.3)
	5-ASA + anti-TNF-α	1 (7.7)	1 (7.7)	1 (8.3)	1 (8.3)
	5-ASA + immunosuppressors + systemic steroids	1 (7.7)	1 (7.7)	1 (8.3)	1 (8.3)
	5-ASA + imunossupressors + anti-TNF-α	1 (7.7)	1 (7.7)	1 (8,3)	1 (8.3)
Disease activity	Remission	13 (100)	13 (100)	9 (75)	8 (66.7)
	Mild	0	0	3 (25)	3 (25)
	Moderate	0	0	0	1 (8.3)

5-ASA: 5-aminosalicylic acid; CD: Crohn’s disease; TNF: tumor necrosis factor; UC: ulcerative colitis.

**Table 3 nutrients-13-00642-t003:** Anthropometric at study initiation (M1) and after two months (M3).

Anthropometric at M1 and after M3	Cases with Intervention		*p* Value	Cases without Intervention		*p* Value	Control with Intervention		*p* Value	Control without Intervention		*p* Value
	Average ± SD			Average ± SD			Average ± SD			Average ± SD		
	M1	M3		M1	M3		M1	M3		M1	M3	
Weight (kg)	74.5 ± 12.2	72.6 ± 11.2	0.032	82.3 ± 8.1	81.6 ± 7.6	0.165	74.9 ± 5.2	72.9 ± 4.9	0.001	75.71 ± 12.0	74.52 ± 11.6	0.071
BMI (kg/m^2^)	24.9 ± 3.1	24.4 ± 2.8	0.039	26.9 ± 2.5	26.8 ± 2.5	0.626	24.9 ± 1.4	24.1 ± 1.3	0.001	26.3 ± 2.6	25.9 ± 2.5	0.08
Waist circumference (cm)	87.0 ± 11.9	83.6 ± 10.4	0.005	96.1 ± 7.4	95.1 ± 6.7	0.102	88.8 ± 6.2	86.0 ± 5.0	0.004	88.2 ± 11.5	85.5 ± 11.2	0.01
% Body fat	21.6 ± 4.6	21.1 ± 5.7	0.456	26 ± 7.9	25.4 ± 6.9	0.392	19.0 ± 4.7	18.2 ± 4.4	0.216	26.2 ± 6.2	25.7 ± 6.1	0.409
Lean body mass (kg)	58.3 ± 9.4	57.2 ± 9.2	0.023	60.8 ± 8.5	60.9 ± 8.1	0.84	60.7 ± 3.6	59.5 ± 3.7	0.001	55.6 ± 9.5	55.3 ± 9.7	0.213

SD: standard deviation; BMI: body mass index.

**Table 4 nutrients-13-00642-t004:** Macro and micronutrients intake between first nutrition appointment (M1) and final evaluation after two months (M3) of different groups.

Macro and Micronutrients Intake between M1 and M3	Cases with Intervention		*p* Value	Cases without Intervention		*p* Value	Control with Intervention		*p* Value	Control without Intervention		*p* Value
	Average ± SD			Average ± SD			Average ± SD			Average ± SD		
	M1	M3		M1	M3		M1	M3		M1	M3	
Energy	1989 ± 524	1855 ± 546	0.18	1866 ± 548	1851 ± 430	0.95	2473 ± 1003	2213 ± 686	0.15	1883 ± 553	1836 ± 509	0.86
(kcal/day)												
Energy	28 ± 9	26 ± 10	0.51	23 ± 6	23 ± 5	0.59	33 ± 14	30 ± 9	0.12	26 ± 10	25 ± 9	0.6
(kcal/kg body weight)												
Proteins	100.1 ± 30.5	93.7 ± 28.4	0.44	86.8 ± 31.9	101.6 ± 28.5	0.27	121.3 ± 38.4	112.2 ± 1.7	0.27	106.4 ± 30.0	105.6 ± 29.9	0.77
g/day												
Proteins	1.4 ± 0.5	1.3 ± 0.5	0.64	1.0 ± 0.3	1.2 ± 0.32	0.21	1.6 ± 0.6	1.5 ± 0.2	0.31	1.45 ± 0.5	1.5 ± 0.54	0.7
g/kg												
Lipids (g/day)	71.3 ± 28.8	55.5 ± 30.7	0.05	62.8 ± 33.5	60.8 ± 26.8	0.66	86.6 ± 45.6	72.8 ± 43.4	0.2	61.4 ± 28.1	60.7 ± 28.9	0.81
Carbohydrates (g/day)	223.7 ± 73.5	240.9 ± 64.2	0.39	230.6 ± 84.9	218.6 ± 49.1	0.63	256.4 ± 122.1	258.8 ± 91.5	0.85	201.1 ± 66.4	200.9 ± 59.2	0.62
Alcohol (g/day)	6.3 ± 15.7	0.4 ± 1.5	0.14	3.6 ± 6.6	2.7 ± 4.7	0.59	24.8 ± 26.3	4.4 ± 8.5	0.01	11.2 ± 14.8	8.8 ± 14.1	0.31
Fiber (g/day)	21.2 ± 10.3	30.1 ± 12.9	0.01	23.3 ± 9.7	22.2 ± 3.7	0.69	29.9 ± 9.4	30.7 ± 12.3	0.88	23.5 ± 8.8	23.2 ± 6.1	0.69
Vitamin D (μg)	4.0 ± 3.4	5.8 ± 7.9	0.68	3.5 ± 4.3	4.7 ± 4.8	0.25	6.8 ± 6.7	3.6 ± 4.4	0.38	6.6 ± 11.4	7.3 ± 14.1	0.7
Vitamin B12 (μg)	3.9 ± 1.9	3.7 ± 3.1	0.43	3.3 ± 2.2	4.3 ± 2.3	0.09	4.5 ± 2.1	3.7 ± 1.2	0.13	9.3 ± 20.1	3.4 ± 1.3	0.94
Folic acid (μg)	246.7 ± 134.5	266.2 ± 134.5	0.55	252.9 ± 83.4	237.5 ± 77.3	0.59	352.9 ± 209.5	259.4 ± 75.2	0.13	258 ± 111.7	271.3 ± 81.3	0.35
Calcium (mg)	696.6 ± 449.3	912 ± 279.4	0.09	741.5 ± 311	893.1 ± 327.7	0.04	1013.9 ± 404.7	1048 ± 402.1	0.99	924.2 ± 308.8	1020.80 ± 350.3	0.25
Iron (mg)	10.8 ± 4.1	9 ± 3.7	0.02	10.5 ± 3.2	10.2 ± 2.4	0.82	14.9 ± 5.7	11.7 ± 5.8	0.01	11.5 ± 4.4	10.1 ± 2.2	0.33
Zinc (mg)	10.8 ± 4.5	9.6 ± 4.5	0.28	9.6 ± 3.4	11.5 ± 3.8	0.23	14.8 ± 6	14.4 ± 3.5	0.54	11.1 ± 3.3	12.3 ± 3.1	0.17
Tryptophan/60 (mg)	18.1 ± 7.8	155 ± 5.9	0.17	16.3 ± 7	7 ± 15.5	0.23	23 ± 7.5	22 ± 3.3	0.43	20.1 ± 6.3	19.3 ± 4.7	0.96

SD: standard deviation.

**Table 5 nutrients-13-00642-t005:** Analytical parameters comparison from M1 to M3 of different groups.

Analytical Parameters from M1 to M3	IBD Patients with Intervention		*p* Value	IBD Patients without Intervention		*p* Value	Controls with Intervention		*p* Value	Controls without Intervention		*p* Value
	Average ± SD			Average ± SD			Average ± SD			Average ± SD		
	M1	M3		M1	M3		M1	M3		M1	M3	
Iron (µg/dL)	95.1 ± 33.8	113.3 ± 39.8	0.131	106 ± 35.8	98.9 ± 33.3	0.349	103.1 ± 32.9	96.2 ± 25.8	0.382	111.6 ± 28.5	110.4 ± 27.6	0.909
Ferritin (ng/mL)	148.1 ± 171.6	149.9 ± 166.5	0.9	152 ± 145	143.3 ± 114.1	0.721	262 ± 151.6	264.9 ± 146.9	0.279	187.3 ± 181.2	201.4 ± 185.0	0.463
Folic acid (ng/mL)	6.51 ± 3.4	8.1 ± 4.8	0.754	8.5 ± 5.1	9.58 ± 4.9	0.838	6.8 ± 2.3	6.0 ± 1.8	0.039	8.19 ± 4.2	6.8 ± 2.6	0.529
Vitamin B12 (pg/mL)	458.6 ± 115.8	462.0 ± 122.7	0.987	489.9 ± 123.7	410.3 ± 115.8	0.01	500 ± 133.9	463.5 ± 123.9	0.125	487.0 ± 160.6	440.3 ± 142.1	0.023
CRP (mg/L)	5.6 ± 13.7	1.8 ± 2.1	0.142	3.2 ± 2.6	2.6 ± 3.0	0.328	1.8 ± 4.6	0.6 ± 0.7	0.528	1.6 ± 1.3	1.4 ± 1.3	0.789
Vitamin D (ng/mL)	22.9 ± 7.9	33.3 ± 8.8	0.00	19.4 ± 10.5	29.7 ± 10.8	0.004	23.3 ± 6.4	29.7 ± 6.5	0.002	27.2 ± 5.6	35.7 ± 9.7	0.00
Total calcium (mg/dL)	9.6 ± 0.5	9.6 ± 0.5	0.53	9.7 ± 0.3	9.6 ± 0.4	0.607	9.6 ± 0.3	9.6 ± 0.2	0.339	9.7 ± 0.4	9.62 ± 0.4	0.671
Calprotectin fecal (µg/g)	470 ± 954.5	316 ± 764.5	0.47	438.6 ± 462.3	61.6 ± 49.5	0.075	32.9 ± 60.8	28.9 ± 31.9	0.071	24.5 ± 26.2	18.6 ± 37.9	0.456
Zinc (µg/dL)	90.2 ± 18.7	86.5 ± 19.4	0.51	92.6 ± 13.7	86.6 ± 12.1	0.325	101.2 ± 15.7	93.9 ± 11.9	0.148	95.2 ± 16.2	92.6 ± 13.7	0.523
Zonulin (ng/mL × 20)	41.2 ± 10.5	41.3 ± 6.1	0.987	61.0 ± 18.5	53.9 ± 17.6	0.265	41.7 ± 10.6	43.8 ± 5.7	0.515	44.4 ± 9.3	48.9 ± 7.9	0.435

SD: standard deviation.

**Table 6 nutrients-13-00642-t006:** Gastrointestinal symptomatology changes between M1 and M3 of different groups.

Symptoms	Evolution of Symptom Prevalence (%)			
	Cases with the Intervention (%M3–%M1)	Cases without Intervention (%M3–%M1)	Control with Intervention (%M3–%M1)	Control without Intervention (%M3–%M1)
Abdominal pain	−21	−19	+9	+9
Abdominal distension	+21	0	+2	+18
Constipation	+29	+18	−14	−6
Diarrhea	0	−19	−6	−6
Flatulence	0	−9	−2	+21
Heartburn/burning	−15	−9	−7	−21

## Data Availability

The data presented in this study are available on request from the corresponding author.

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
