# Peer review of "Functional Food Components, Intestinal Permeability and Inflammatory Markers in Patients with Inflammatory Bowel Disease"

_nutrients, 2021, doi:10.3390/nu13020642_

Round 1
Reviewer 1 Report
The study is pretty interesting if it would deliver more controlled nutritional plan.
The manuscript can be improved by following points for more priority in the nutritional aspect rather than clinical report.
The main concern
-Intake based on DP was not evenly performed b/w intervention and non-intervention groups, which led to biased conclusion in DP effectiveness. It is very hard to tell the many improvements are associated with DP. It seems to be due to therapeutics.
- More clear explanation on DP and adherence rates in the different group.
- Author should provide more clear evidences or presentation in terms of DP efficacy.
- Graphic presentations in DP-associated results would be recommended.
- In terms of DP efficacy, comparison b/w control, UC, and CD needed to be evaluated.
- Mechanistic explanations on different DP adherence in patients w/ and w/o intervention.
Author Response
Dear Reviewers
Hereby we send the answers and comments to all different questions presented to us.
We must thank for all the effort done in carrying out a careful analysis of our work.
We analyzed all the suggestions and tried to respond to each one.
Conscious that we have a small sample size, with an important limitation (diet compliance) we believe that this work can be useful for future studies that aimed to use diet as co-adjuvant to pharmacological therapy.
We remain available to improve the document if you consider it necessary.
Kind regards
Catarina

Reviewer 2 Report
The article “Functional food components, intestinal permeability and inflammatory markers in patients with Inflammatory Bowel Disease” is interesting. Unfortunately, the number of patients included in the study is very small and no clear-cut conclusions can be drawn from it. Additionally, there are several issues that I think should be addressed prior to publication:
- I think the novelty of the study should be emphasized more. Also, perhaps in the Introduction section, the impact of IBD in terms of incidence, cost, etc and the potential benefit of finding a non-pharmacological method of alleviating some of the symptoms should also be emphasized.
- In the Materials and Method section, I think you should also clearly list the criteria used for diagnosing IBD. Also, why did you only exclude patients with type 1 DM and not patients with type 2 DM?
- How did you decide the number of patients that were to be included in the study? Were there no screening failures in this study (figure 1 suggests that there have been none)...
- The difference between the intervention group and the control group in terms of their diet is somewhat fuzzy. They all received a personalized diet plan, but curcumin was compulsory only in the intervention group? I think the two types of diet should be clearly listed and the differences underlined.
- In Table 1, in the section describing smoking habits you have 3 smokers and 23 non-smokers, which adds up to a total of 26 (when you have 25 participants)
- In Table 3, I noticed that waist circumference in almost 10 larger in IBD-intervention group vs IBD-no intervention group. A similar difference between groups in terms of body fat (21.6 vs 26.7). Were these differences statistically significant? How do you explain these differences?
- Why did you not use a QOL questionnaire or another validated tool to assess gastro-intestinal symptoms?
- Why were ZO values similar in the IBD vs non-IBD group? Should you not have excluded IBD patients with no IP problems since the aim of the study was to assess the effect of a nutritional intervention on IP?
- English needs to be improved – it appears as if the article has been translated from another language and some sentences or words are not translated. Some examples below:
Row 15 – “recent evidence suggests that” instead of “recent evidence suggest that”
Rows 34-36 – “Inflammatory bowel disease (IBD) is characterized by a chronic inflammatory process of the gastrointestinal tract and is marked by a modification of the intestinal barrier and thereby disrupting intestinal permeability” – please rephrase
Rows 50-51 – “(quercetin and curcumin)” instead of “(quercetin e curcumin)”
Rows 51-52 – “prebiotics (beta-glucans and butyrate) and glutamine” instead of “prebiotics (beta-glucans e butyrate) e glutamine”
Rows 144-145 – “Regarding the clinical data, almost 70% of IBD patients IBD and 50% of controls didn’t present comorbidities.” – please re-check
Rows 149-150 – “During the study, 1 participant raised the medication dosage, 6 reduced, 1 stopped corticotherapy and 5 reduced the dosage of 5-ASA). At the end all participants were under some kind of therapeutic” – add parenthesis and re-check
Row 161 – “was observed was in both” instead of “was observed in both”
Row 197 – “These correlations lost their clinical relevance” instead of “These correlations lost their clinically relevance”
Row 216 – “Although a quite high satisfaction regarding to DP satisfaction was referred” – please rephrase
Author Response

(The authors gave the same response as above.)

Reviewer 3 Report
This is an interesting topic, and the authors have put much effort into the investigation of dietary aspects of chronic inflammatory bowel diseases. Moreover, they did this in a complicated patient group, with poor compliance and medical adherence beyond taking pills. This should be explained and pointed out specifically. Nevertheless, there are principal considerations concerning the study.
Major:
- L 117: The reviewer wonders, why the intervention duration was only 8 weeks.
- Was there any power analysis to define the group size, particularly with respect to an anticipation of drop out numbers. May be 90% drop outs (L. 208f) were unexpected, but it generally has to be included in study planning. As there was nearly no compliance, we have an “ intention to treat” (ITT) study with a non-compliant patient group, rather than raising effectivity data of an intervention. There is no separate data set for the compliant participants.
- L. 102ff: There is no methods description how intake was assessed (3d-questionnaire or other). The authors refer to tryptophan(e), Zinc, Vit. D etc, but only control for Vit. D. Define “Mediterranean Diet”. What means “encouragement”?
- Many important components that have not been analyzed, can be analyzed or implemented in calculations using a questionnaire. Gliadin for example can be assessed when knowing how much wheat and other grains were ingested. Ethanol, additives and others can be calculated from questionnaire data and EU food ingredients lists in commercial food products as well. There is no assessment of the reduction of "fast food" as important factors in IBD.
- How was Vit. D supplementation done? Was it only controlled at visit 3? Table 5 shows that according to standard deviation values about 80% of patients were below 30ng/mL at the beginning, but still about 50% after 2 months. Hence, Vit. D was not effectively corrected. Partial correction of a confounder during, rather than prior to a trial, may compromise the results.
- Line 71: As hepatic steatosis (NAFLD) is a frequent characteristic in IBD, possibly resulting from intestinal barrier disruption, the reviewer doesn’t consider this as adequate. Intestinal inflammation, together with metabolic factors, contributes to the pathogenesis of IBD-associated NAFLD. Moreover, how was steatosis confirmed? A 7% fat content of livers is not easily discovered by routine ultrasound imaging. It is more frequent at BMI >25kg/m2, probably in both controls and IBD patients.
- L. 88: Why were alpha-1-antitrypsin, sIG-A and zonulin not measured in stool?
Minor:
- L. 18: 47%, not 50%
- L. 36: Is permeability or the barrier disrupted?
- L 43. What is seric? Didn’t find the word in dictionary. Where was concentration measured. If in plasma, why not in stool?
- L. 44: double space between ing & a. Similarly in many places of the manuscript as well.
- L. 51: What does “e” mean. Is it a Spanish word accidentally remaining in an English text? So in other places as well!
- L. 65: informed >written< consent?
- L. 80: How was randomization performed?
- L 120ff: Fig 1: Fonts are too small for readability.
- L 112: Smoking is an important confounder!
- Table 1-5: Data frequently don’t seem to be normally distributed. Provide medians, IQR and full range. Explain all abbreviations of a table in the legend
- Table 3: Include units!
- L. 152, 1st sentence: Unclear sentence. Remission at the end or before start?
- L. 163ff: 1st sentence: How was that determined? Please define the methods used precisely.
- L. 166f: This is not a complete sentence.
- L. 169: >was< rather than >were<.
- Table 4: It might be useful to create 2 parts of table, with total intake (A) and intake/kg BW (B)
- Table 6: The data should be presented in more detail, with median values/IQR and p-values
- L. 186ff: Table 5 (and others) The data don’t look like being normally distributed. Use medians and IQR.
- L. 201f: What does this mean, if there is no p<0.05 after test? Either the declaration is wrong or the statistics. Show values in a table.
- L. 230: What does that mean? It is not possible to analyse the samples, befor they have been taken.
- L. 282: For limitations use new para with Caption.
- L 297: >Therefore< rather than >Wherefore<?
- L. 309: This statement is not part of the discussion.
- General: Check for grammar.
Author Response

(The authors gave the same response as above.)

Round 2
Reviewer 1 Report
- Mostly addressed except for DP efficacy which can be mentioned in the text.
- Typo and space issue: all units should be spaced with the number except for % and temperature. many typos in number eg> 8,75?-->8.75, 2,5---->2.5....et al.
Author Response
R1
1. Mostly addressed except for DP efficacy which can be mentioned in the text.
A: Thank you, was added this information (L.54-89)
Typo and space issue: all units should be spaced with the number except for % and temperature. many typos in number eg> 8,75?-->8.75, 2,5---->2.5....et al.
A: Thank you, it was added edited.
R3
The authors have appropriately answered to the objections raised ba the reviewer. There is only 1 minor comment on table 1: the differences between means and medians indicate that values are not normally distributed. Hence, while standard deviation of the mean is a single value, the interquartile range of median values (50% below, 50% above) is NEVER a single value. Instead it is one smaller than median (25% are below that value), and one above the median (25% are above that value). Please correct!
A: Thank you, it was added edited.
Reviewer 2 Report
the authors have answered the queries
Author Response

(The authors gave the same response as above.)

Reviewer 3 Report
The authors have appropriately answered to the objections raised ba the reviewer. There is only 1 minor comment on table 1: the differences between means and medians indicate that values are not normally distributed. Hence, while standard deviation of the mean is a single value, the interquartile range of median values (50% below, 50% above) is NEVER a single value. Instead it is one smaller than median (25% are below that value), and one above the median (25% are above that value). Please correct!
Author Response

(The authors gave the same response as above.)
